# *Anaplasma phagocytophilum* Transmission Activates Immune Pathways While Repressing Wound Healing in the Skin

**DOI:** 10.3390/life12121965

**Published:** 2022-11-24

**Authors:** Jacob Underwood, Cristina Harvey, Elizabeth Lohstroh, Branden Pierce, Cross Chambers, Stephanie Guzman Valencia, Adela S. Oliva Chávez

**Affiliations:** 1Department of Entomology, Texas A&M University, College Station, TX 77845, USA; 2Navy Entomology Center of Excellence, United States Navy, Jacksonville, FL 32212, USA

**Keywords:** transmission, ticks, innate immune responses, metalloproteinases, extracellular matrix, neutrophils, cytokines, intracellular bacteria

## Abstract

*Anaplasma phagocytophilum,* the causative agent of human granulocytic anaplasmosis (HGA), is an obligate intracellular bacterium transmitted by the bite of black-legged ticks, *Ixodes scapularis*. The main host cells in vertebrates are neutrophils. However, the first site of entry is in the skin during tick feeding. Given that the initial responses within skin are a crucial determinant of disease outcome in vector-borne diseases, we used a non-biased approach to characterize the transcriptional changes that take place at the bite during *I. scapularis* feeding and *A. phagocytophilum* transmission. Experimentally infected ticks were allowed to feed for 3 days on C57BL/6J mice to allow bacterial transmission and establishment. Skin biopsies were taken from the attachment site of uninfected ticks and *A. phagocytophilum*-infected ticks. Skin without ticks (intact skin) was used as baseline. RNA was isolated and sequenced using next-generation sequencing (NGS). The differentially expressed genes were used to identify over-represented pathways by gene ontology (GO) and pathway enrichment (PE). *Anaplasma phagocytophilum* transmission resulted in the activation of interferon signaling and neutrophil chemotaxis pathways in the skin. Interestingly, it also led to the downregulation of genes encoding extracellular matrix (ECM) components, and upregulation of metalloproteinases, suggesting that *A. phagocytophilum* delays wound healing responses and may increase vascular permeability at the bite site.

## 1. Introduction

The black-legged tick, *Ixodes scapularis*, has been spreading in recent years to more locations across the northeastern and midwestern United States. It is a competent vector of seven different pathogens, known to cause illness in humans, including *Anaplasma phagocytophilum* [1]. *Anaplasma phagocytophilum* is the causative agent of human granulocytic anaplasmosis (HGA), formerly known as human granulocytic ehrlichiosis (HGE), which was discovered infecting a cluster of men in the upper Midwest in 1994 [2]. Later, similar cases were reported in other parts of the U.S. where Lyme disease is prevalent [3]. Most recently, 5655 cases were reported to the CDC in 2019, making HGA the second most common tick-borne disease in the U.S. [4]. In most cases, the illness is self-limiting, and patients will recover with or without antibiotic treatment. Symptoms will typically develop 1–2 weeks following a bite by an infected tick, can lead to hospitalization in around 36% of confirmed cases, and presents mortality rates of around 0.2–1% [5].

*Anaplasma phagocytophilum* is a Gram-negative bacterium of small size (0.4 to 1.5 μm) that replicates inside neutrophils within small vacuoles termed “morulae” [6]. This bacterium lacks several of the enzymes involved in peptidoglycan and lipopolysaccharide biosynthesis [6,7], which are two pathogen-associated molecular patterns (PAMPs) commonly recognized by vertebrate immune responses. Nevertheless, stimulation of peripheral blood leukocytes from healthy donors with cell free bacteria or recombinant *A. phagocytophilum* outer protein P44 (rP44) induces the expression of proinflammatory cytokines [8]. Similarly, high levels of proinflammatory cytokines, such as interferon (Ifn)-γ, interleukin (IL)-12p70, and IL-10, are detected in HGA patients [9]. Further, the elevated levels of these cytokines appear to be associated with the severity and pathology of the disease in humans and murine models [9,10,11]. Ifn-γ is involved in the control of *A. phagocytophilum* infection in mice [10,11,12] and during the in vitro infection of Hoxb8 neutrophils [13]. In vivo experiments using murine models indicate that secretion of Inf-γ correlates with Stat1 phosphorylation [14]. Stat1 knock-out in mice leads to increased bacterial loads, severe disease pathology, increased spleen size, higher cytokines/chemokines levels in plasma, and reduced iNOS induction [15]. These studies were focused on systemic immune responses, and little is known about the immune events that take place during *A. phagocytophilum* transmission at the bite site.

Despite the importance of the skin as the site of entry and establishment of *A. phagocytophilum*, only a few experiments have been conducted to understand the events that occur at the bite site. In 2010, Granquist et al. [16] described the site of *A. phagocytophilum* transmission in lambs naturally infested with *Ixodes ricinus* ticks. Histological inspection of the bite sites showed evidence of inflammation, accumulation of immune cells such as neutrophils, macrophages, and other mononuclear cells, and the deterioration of the collagen matrix. Bacteria were associated with neutrophils and macrophages. However, the time of feeding for each tick could not be determined since animals were naturally infested. A later study similarly reported the presence of *A. phagocytophilum* within neutrophils in the skin of experimentally infected sheep later infested with ticks [17]. This study also reported significantly higher numbers of neutrophils in the skin of the experimentally infected sheep independent of the presence of feeding ticks, suggesting that *A. phagocytophilum* infection increases migration of neutrophils to the skin. Although this study considered the changes in expression of eight immune-related genes, the responses by each animal were variable and no conclusions on the activation of immune signaling pathways could be drawn. Thus, an important knowledge gap exists on the signaling and immunological events that take place during transmission of *A. phagocytophilum* by ticks.

The present study explores the gene expression changes that occur at the bite site during the transmission of *A. phagocytophilum.* We describe an upregulation in genes involved in Ifn-γ signaling, defense responses to viruses, neutrophil chemotaxis, and interleukin-1 responses. Interestingly, *A. phagocytophilum* transmission appears to decrease the expression of genes implicated in extracellular matrix (ECM) organization and wound healing responses. Given that the skin is the initial site of *A. phagocytophilum* establishment, an understanding of the immunological events that take place during initial infection will help us uncover potential signaling pathways and immune responses that can be exploited to stop infection.

## 2. Materials and Methods

### 2.1. Anaplasma Phagocytophilum Culture

HL60 cells (CCL-240™) were obtained from ATCC (Manassas, VA, USA). HL60 cell cultures were maintained in RPMI media (Corning, Manassas, VA, USA) supplemented with 10% fetal bovine serum (Gibco, Whaltham, MA, USA), 1% Glutamax (Gibco, Whaltham, MA, USA), and 1% amphotericin B (Corning, Manassas, VA, USA) and incubated at 37 °C with 5% CO_2_, as previously described [18]. Cells were maintained until the cell density became optimal for passage or for infection with *A. phagocytophilum* (~1 to 5 × 10^5^ cells/mL). Cells were passaged as follows: 2 mL of cell culture was transferred into a 25 cm^2^ tissue culture flask (Fisher Scientific, Pittsburgh, PA, USA) and supplemented with 18 mL of freshly prepared culture media. This was repeated every 3 to 5 days, until cells reached passage 10 when they were discarded, and a new culture was recovered from liquid nitrogen (LN_2_).

For infection with *A. phagocytophilum*, 2 mL of uninfected HL60 cell cultures (at approximately 5 × 10^5^ cells/mL) was inoculated with 500 μL of *A. phagocytophilum*-infected HL60 cells (at approximately 2 × 10^5^ cells/mL, with ~90% infection). *Anaplasma phagocytophilum* was cultured in HL60 cells for up to 5 days. Infections were monitored by placing 1 mL of the suspended cell culture onto a microscope slide, and spinning with a CytoSpin 4 (Thermo Scientific, Whaltham, MA, USA) at 800× *g* for 5 min. The infected cells were stained using the Richard-Allan Scientific™ Three-Step Stain Kit (Thermo Scientific, Whaltham, MA, USA), according to manufacturer’s specifications. The morulae within cells were observed by light microscopy with an Olympus model BX43F (Shinjuku City, Tokyo, Japan). Infections were passaged when the percentage of infection was greater than 90%, determined by counting 100 HL60 cells with observable morulae. Bacterial cultures were maintained for up to 5 passages before freezing or infecting mice, using the procedures described below.

### 2.2. Mice Infections

C3H/HeJ male mice of 6 weeks of age (The Jackson Laboratory, Bar Harbor, ME, USA) were used for pathogen acquisition due to their high susceptibility to infection from Gram-negative bacteria, including to *A. phagocytophilum* infection [19]. The susceptibility of this mouse strain to Gram-negative bacteria is associated with a mutation in the cytoplasmic domain of Toll-like receptor 4 (TLR4) [20] and has shown impaired inflammatory and innate immune responses under several conditions [21,22]. Mice were injected intraperitoneally (i.p.) with 100 μL *A. phagocytophilum*-infected HL60 containing 1 × 10^7^ bacteria, using 27-gauge needles (Figure 1a). Cells were spun down at 300× *g* for 10 min, culture media was removed, and the cells were suspended in 1× PBS. The number of bacteria was estimated using the previously described formula [23]. Control mice received an injection of 100 μL 1× PBS.

To confirm infections, cheek bleeds were performed on the mice on days 3, 5, and 7 post-infection (p.i.), collecting 20 to 100 μL of blood in microvettes 500 K3E (Sarstedt, Nümbercht, Germany) after anesthesia with 1.25% to 2% isoflurane (Figure 1a). Blood was used for DNA extraction with the DNeasy Blood & Tissue Kit (QIAGEN, Hilden, Germany), following the manufacturer’s instructions. DNA quantity and quality was assessed using a NanoQuant Infinite M200 Pro (Tecan, Switzerland). PCR amplification of mouse actin was performed to confirm the absence of contaminants and PCR inhibitors. PCR analyses of the *A. phagocytophilum rpoB* and the *16s* rRNA genes were performed on the blood to confirm infection with *A. phagocytophilum*. PCRs were prepared using GoTaq Flexi DNA polymerase (Promega, Madison, WI, USA). Amplification was completed using the following PCR cycling conditions: 1 denaturing cycle for 3 min at 95 °C, followed by 34 cycles of 1-min denaturation at 95 °C, 1 min at the annealing temperature (Table 1) and an extension of 72 °C for 30 s. A final extension step of 5 min at 72 °C was performed. Predicted product sizes are displayed in Table 1. PCR product sizes were confirmed by gel electrophoresis and compared with a 100 bp ladder (NEB, Ipswich, MA, USA). Gels were visualized with an iBright FL1500 Imaging System (Thermo Scientific, Whaltham, MA, USA) to confirm the infection of mice (Appendix A).

### 2.3. Tick Infestations

For *A. phagocytophilum* tick infections, 200 larval *I. scapularis* ticks were placed on mice after confirming infection in the blood (Figure 1a). Mice were separated into individual mesh bottom cages placed above a water trap to collect the fed ticks. Mice were anesthetized for 30 min with 1.25% to 2% isoflurane to allow the larvae to attach. Engorged ticks were collected from the water baths after 3, 4 and 5 days of feeding (Figure 1a). The mice were euthanized with CO_2_, followed by cervical fracture and heart puncture exsanguination. The collected ticks were washed in 2% bleach and autoclaved water, placed into groups of 25, and allowed to molt into nymphs.

DNA was purified from 5 pooled nymphs from each group of infected and control ticks to confirm infection or lack of. Ticks were placed at −80 °C for 1 h and DNA was isolated using the Quick-DNA/RNA Miniprep kit (Zymo, Irvine, CA, USA), according to the manufacturer’s instructions. DNA quantity and quality were tested as described above, and a PCR on the *I. scapularis* actin gene was performed to determine the presence of PCR inhibitors. *Anaplasma phagocytophilum* infection (or lack of) was confirmed by PCR amplification of the *rpoB* and *p44* genes (Table 1; Figure 1a). Amplification of the PCR products was carried out using similar cycling conditions as described above. Predicted product sizes are displayed in Table 1. Positive and negative PCRs were confirmed by gel electrophoresis as described above. Additionally, the relative levels of bacterial infection were assessed by qPCR, using the Δ*Ct* value of *A. phagocytophilum p44* normalized by tick *actin* with the following formula:$$\Delta Ct=2^{-\left(ct\;Anaplasma\;p44-ct\;tick\;actin\right)}$$

qPCRs were performed using PowerUp™ SYBR™ Green Master Mix (Applied Biosystems, Whaltham, MA, USA), using same primers as for PCR analysis (Table 1). Amplification, melt curves, and data were analyzed with CFX Maestro Software (Bio-Rad, Hercules, CA, USA).

Due to the potential effect of the mutation of C3H/HeJ mice *tlr4* in the local immune responses to *A. phagocytophilum* transmission, we decided to use a different mouse strain. The C57BL/6J mouse strain has previously been used for the study of murine systemic immune responses during *A. phagocytophilum* infection and the role of IFN-γ/STAT1 [11], therefore, we used this same strain to define local immune responses to bacterial transmission. *A. phagocytophilum*-infected and -uninfected nymphs were used to infest 6 week old C57BL/6J male mice (The Jackson Laboratory Bar Harbor, ME, USA). Twenty-five *I. scapularis* nymphs were placed on each mouse. Mice were anesthetized as previously described for the larvae (Figure 1b). The ticks were allowed to feed for 3 days to allow 24 h post-transmission of *A. phagocytophilum* [28]. After this time, the mice were euthanized with CO₂, followed by cervical fracture. Three (3) mm skin biopsies were taken of the bite sites, utilizing Integra disposable biopsy punches (Militex, Saint-Pries, France) for RNAseq, and 5 mm skin biopsies were taken for qRT-PCR (Figure 1b). Skin samples far from where the ticks were located were taken to determine the gene expression in intact skin (baseline). The partially engorged nymphs were collected for excision of their midguts and salivary glands. The skin was placed in 500 μL RNALater (Invitrogen, Carlsbad, CA, USA).

### 2.4. RNA-Seq and Pathway Analysis of Skin Biopsies

RNA extraction, library preparations, sequencing reactions and bioinformatic analysis were conducted at GENEWIZ, LLC. (South Plainfield, NJ, USA) as follows: total RNA was extracted using Qiagen RNeasy Plus Universal mini kit following the manufacturer’s instructions (Qiagen, Hilden, Germany). The quantity and quality of the RNA samples were assessed using a Qubit 2.0 Fluorometer (Life Technologies, Carlsbad, CA, USA) and an Agilent TapeStation 4200 (Agilent Technologies, Palo Alto, CA, USA), respectively.

The RNA libraries were prepared using the NEBNext Ultra II RNA Library Prep Kit (NEB, Ipswich, MA, USA), following the manufacturer’s instructions for Illumina. Briefly, mRNAs were first enriched with Oligo(dT) beads, followed by fragmentation for 15 min at 94 °C and cDNA synthesis. cDNA was adenylated at 3′ends and end repaired. Universal adapters were ligated to cDNA fragments, followed by index addition and library enrichment by limited-cycle PCR. The quality of the libraries was validated on the Agilent TapeStation (Agilent Technologies, Palo Alto, CA, USA). Libraries were quantified with a Qubit 2.0 Fluorometer (Invitrogen, Carlsbad, CA, USA) and quantitative PCR (KAPA Biosystems, Wilmington, MA, USA).

The libraries were sequenced on an Illumina HiSeq instrument (4000) according to the manufacturer’s instructions, using a 2 × 150 bp paired end (PE) configuration. HiSeq Control Software (HCS) v2.0.12 was used for image analysis and base calling. After investigating the quality of the raw data, possible adapter sequences and nucleotides with poor quality were trimmed. The trimmed reads were mapped to the reference genome, using STAR aligner v.2.5.2b. Unique gene hit counts were calculated using featureCounts from Subread package v.1.5.2. Only unique reads that fell within exon regions were counted. The complete sequencing data were deposited on NCBI, and accession numbers can be found in the data availability section.

Upregulated and downregulated genes with adjusted *p*-value (padj) < 0.05 and log2 fold changes of 1 or more (−1 or less for downregulated genes) were used to identify pathway over-representation using Reactome (https://reactome.org/, accessed on 19 February 2022). Only pathways with *p* < 0.05 and false discovery rate (FDR) < 0.05 were considered as over-represented.

### 2.5. qRT-PCR of Skin Biopsies

RNA was extracted from the skin using TRIZOL (Invitrogen, Whaltham, MA, USA) according to the manufacturer’s specifications with small modifications. Briefly, the RNALater was washed off the tissues with 1× phosphate buffered saline (PBS), and skin samples were quickly flash froze with LN_2_. The frozen tissue was homogenized with a mortar and pestle. One (1) mL of TRIZOL was added to the tissue. The aqueous phase was taken and mixed 1:1 with 70% ethanol. RNA was then isolated using PureLink™ RNA Mini Kit (Ambion, Carlsbad, CA, USA), according to the manufacturer’s indications. RNA was quantified and quality was assessed using a NanoQuant Infinite M200 Pro (Tecan, Switzerland). RNA (100 ng) was used to synthesize cDNA with the Verso cDNA Synthesis Kit (Thermo Scientific, Whaltham, MA, USA). qPCRs were performed using PowerUp™ SYBR™ Green Master Mix (Applied Biosystems, Whaltham, MA, USA). Interferon gamma (*Ifn-g*), interleukin 1β (*Il1b*), interferon regulatory factor 1 (*Irf1*), S100 calcium binding protein A8 (*S100a8*), Aggrecan (*Acan*) and Matrilin 3 (*matn*3) were amplified using the primers described in Table 1. The amplifications were performed in a CFX Opus 96 Real-Time PCR instrument (Bio-Rad, Hercules, CA, USA). Amplification, melt curves, and data were analyzed with CFX Maestro Software (Bio-Rad, Hercules, CA, USA). The relative differences were calculated using the ΔΔCt method [29] as follows:$$\Delta \Delta CT=\frac{\left(2^{-\left(Ct\;gene\;of\;interest\;treatment-Ct\;actin\;treatment\right)}\right)}{\left(2^{-\left(Ct\;gene\;of\;interest\;intact\;skin-Ct\;actin\;intact\;skin\right)}\right)}$$

The expression of the genes was normalized to mouse *actin* (Table 1). The expression of immune genes in skin samples at the bite site of uninfected and infected ticks was normalized to that of the intact skin (baselines). Statistical differences in gene expression between conditions was evaluated using an unpaired two-tailed *t*-test with GraphPad Prism 9.3.1 (GraphPad Software, San Diego, CA, USA). Outliers were determined using GraphPad 9.3.1.

The specificity of the primers was confirmed by RT-PCR of RNA isolated from mouse hearts followed by Sanger sequencing (Etonbio, Union, NJ, USA). The forward and reverse sequences were assembled with Geneious Prime (Biomatters, Inc., San Diego, CA, USA) and identity was confirmed through BLAST (NCBI). Accession numbers for the sequences amplified by the primers used here in can be found in the data availability statement.

## 3. Results

### 3.1. Tick Feeding Induces the Expression of Neutrophil Chemotaxis and Inflammatory Responses in the Skin

Pathogen-free certified larvae fed upon naïve mice until repletion. Following molting, the absence of *A. phagocytophilum* infection in control nymphs was confirmed by DNA extraction, followed by negative amplification of p44 and RpoB genes from *A. phagocytophilum* (Appendix A). Uninfected (control) nymphs fed upon C57BL/6J for 3 days, and 2 mm skin biopsy punches were taken from the bite site. RNAseq and differential gene expression (DEG) analyses were performed on RNA isolated from the skin samples. A total of 1213 genes were differentially regulated upon tick feeding, when compared with the intact skin (Table 2). A total of 797 genes were upregulated and 416 were downregulated during tick feeding. Principal component analysis (PCA) demonstrated a clear separation of the transcriptomic profiles between the intact skin and uninfected tick bite sites (Figure 2a). Volcano plot analysis of intact skin samples versus uninfected tick bite sites showed the upregulation of several immune-related genes (including genes encoding chemokines and chemoattractant Saa3), and the downregulation of iron homeostasis genes (such as Hamp2) and transcription factors (such as RorC) (Figure 2b).

Heatmap analysis of the 30 most differentially regulated genes resulted in five gene clusters of co-regulated gene (Figure 2c). Cluster 1 included keratine-encoding genes important in epithelial cell integrity in the skin (*krt16* and *krt6b*); enzymes involved in phospholipid and heme metabolism, and glycolysis (*Plbd1*, *Hmox1*, and *Eno1*); and to signaling proteins, including Arrdc3 that allows G protein signaling, and Saa3. Cluster 2 comprised several lectins, such as *Chil3*, *Sell*, and *Clec4d*; the glycoprotein *Cd300lf*; and *2610528a11Rik*, a gene encoding G protein-coupled receptor 15 ligand induced during several inflammatory conditions in the skin [30]. Cluster 3 contained a gene without known function (*Gm45819*) and a circadian transcriptional repressor (*Cry1*). Cluster 4 was formed by two chemokines, one involved in macrophage attraction (*Ccl7*), and *Ccl2* that is a chemoattractant for monocytes and basophils. It also contained a lipocalin (*Lcn2*), a metalloproteases inhibitor (*Timp1*), and a proteoglycan (*Prg4*). Cluster 5 contained two genes involved in chromosome segregation (*Kif22* and *Nuf2*), immune response associated genes (*Slfn4*, *S100a9*, *S100a8*, and *Il1b*), epidermal maintenance and development (Stfa3 and Stfa1), and other genes (*Slc15a3* and *Adamts4*). All clusters showed upregulation of the co-regulated genes, with the exception of Cluster 3 that showed higher levels of expression in the intact skin (baseline; Figure 2c). Enrichment analysis based on gene ontology (GO) (Appendix A) showed an over-representation of genes associated with immune responses, neutrophil chemotaxis, cell division, and chromosome segregation. Pathway enrichment (PE) analysis (Appendix A) of upregulated genes demonstrated enrichment of interleukin-10 signaling associated genes, neutrophil degranulation, Th2 signaling, and mitotic spindle, potentially indicating the division of keratynocytes in response to the inflammation and damage associated with the tick bite as previously reported in mouse biopsies [25]. Downregulated genes did not show any significant enrichment in pathways (Appendix A).

### 3.2. Anaplasma Phagocytophilum Transmission Induces the Upregulation of Interferon Signaling Genes

*Ixodes scapularis* larvae fed upon *A. phagocytophilum* PCR-positive mice (Appendix A). Ticks were allowed to molt and DNA was extracted from five nymphs from each group to confirm infection status. *Anaplasma phagocytophilum*-positive nymphs (Appendix A) with varying relative levels of bacterial infection (Appendix A) were infested onto naïve C57BL/6J mice for 3 days. Skin samples were taken from the bite site and RNA was isolated for RNAseq analysis. DEG analysis identified a total of 2559 DEGs between the intact skin and skin samples from the bite site of *Anaplasma phagocytophilum*-infected ticks, including 1417 upregulated and 1142 downregulated genes. Comparatively, 622 DEGs were identified when uninfected tick and infected tick bite sites were compared. This encompassed 476 upregulated and 146 downregulated genes, indicating a synergistic effect between *A. phagocytophilum* and the tick (Table 2). PCA showed the separation of control samples and samples from the bite site of *A. phagocytophilum*-infected ticks and of uninfected ticks. Similar separation was observed between intact skin and infected ticks’ bite sites. In both cases, the presence of one outlier from the *A. phagocytophilum* ticks was detected (Figure 3a and Appendix A). Nevertheless, this sample showed similar co-regulatory gene expression as the other samples (Figure 3c and Appendix A). Anaplasma4, on the other hand, showed a slightly lower upregulation of immune genes when compared with skin samples from bite sites of uninfected ticks and intact skin (Figure 3c and Appendix A; Appendix A). This might be due to lower levels of infection in the tick or due to a delayed attachment of the tick leading to less time for *A. phagocytophilum* transmission, although this is speculative since we did not test bacterial numbers in the skin.

Analysis of the gene expression using volcano plots showed several immune-related genes that are significantly upregulated during transmission of *A. phagocytophilum* when compared with uninfected ticks’ bite sites (Figure 3b), including the gene encoding tumor necrosis factor (tnf), and chemokines ccl4 and cxcl10, which are attractants of macrophages, natural killer cells, and T cells. Interestingly, genes involved in extracellular matrix formation were downregulated; for example, the gene encoding matn3. Heatmap analysis identified six clusters of co-regulated genes (Figure 3c). The first cluster upregulated in *Anaplasma transmission* skin samples, included the genes encoding for the interferon gamma inducible GTPase Ifgga3 protein (*Gm4841*), lymphocyte antigen 6c2 (*Ly6c2*; which acts as an acetylcholine receptor), the endoplasmic reticulum/Golgi membrane-spanning 4-domains subfamily A, member 4C (*Ms4a4c*), and the ubiquitin-like modifier protein ISG15 (*Isg15*). The second cluster contained two interferon-induced genes (*Ifi213* and *Oas3*), regulators of Th1 cytokine secretion (*Phf11b* and *Phf11d*), and an enzyme involved in arginine biosynthesis (*Ass1*), which showed upregulation in the skin samples from infected ticks bite sites.

A third cluster of upregulated genes involved *Slfn4*, the gene encoding an interferon regulatory factor 7 (*Irf7*), and the interferon-induced gene *Oasl2*. Two genes encoding G receptor interacting proteins were also found in this group (*Rgs5* and *Slc9a3r1*). The fourth cluster only contained two genes slightly upregulated during *A. phagocytophilum* transmission: an interferon activated gene (*Ifi214*) and a pseudogene (*Gm654*). The fifth cluster contained the highest number of genes and included several immune-related genes (*Gzmb*, *Gm12185*, *Trim30b*, *Ccl4*, and *Tnf*) and two lipid transport proteins (*Apol9b* and *Apol9a*). The last cluster included genes encoding two glycoproteins (*Cd300lf* and *Clec4e*) and an interferon-induced protein (*Ifi44*), which are involved in innate immune responses (Figure 3c). Several of the immune-related clusters upregulated in *A. phagocytophilum* transmission sites versus controls were also observed when compared with intact skin (baseline; Appendix A).

Gene ontology (GO) enrichment analysis of the differentially regulated genes during A. phagocytophilum transmission identified the enrichment of genes involved in the cellular response to interferon β (Ifn-β) and Ifn-γ (Figure 4). Enrichment of interferon signaling was also detected using PE analysis of the significantly upregulated genes (Table 3 and Appendix A). Interestingly, antiviral mechanisms also showed enrichment in GO and pathway analysis, likely due to the intracellular nature of *A. phagocytophilum* infection. Other interleukin signaling pathways (IL-1, IL-10, and IL-12) were also observed in the enrichment analysis. Th2 responses were enriched to a lesser degree and likely represented the responses to the tick feeding, as these pathways were significantly over-represented in uninfected tick bite sites, unlike interferon signaling that was not found to be upregulated during uninfected tick feeding (Appendix A). This suggests that *A. phagocytophilum* transmission leads to the activation of Th1 signaling pathways in the skin.

Pathway enrichment (PE) analysis of downregulated genes during *A. phagocytophilum* showed an over-representation of pathways involved in ECM integrity, including ECM organization, fibrils formation, and several collagen organization enzymes (Table 4). Some genes involved in these pathways were downregulated in uninfected tick bite sites (Appendix A). Nevertheless, pathways involved in ECM integrity were not over-represented in uninfected tick bite sites (Appendix A). Curiously, although the upregulation of interferon and interleukin signaling pathways were detected during *A. phagocytophilum* transmission when compared with intact skin (Appendix A), the ECM organization pathways were not over-represented in downregulated genes (Appendix A).

### 3.3. Differentially Expressed Genes (DEGs) Stimulated during Tick Feeding and A. Phagocytophilum Transmission

To determine genes and pathways that are affected by both the tick feeding and *A. phagocytophilum* transmission, the DEGs that are upregulated and downregulated during the bite of uninfected and infected ticks, when compared with intact skin expression, were identified (Appendix A). Pathway enrichment (PE) of genes upregulated in both conditions showed an over-representation of interleukin-10 signaling, neutrophil degranulation, chemokine signaling, Th2 cytokine signaling, and other pathways enriched during tick feeding compared with intact skin expression (Table 5 and Appendix A), corroborating that the enrichment of these pathways during *A. phagocytophilum* transmission is the result of the synergetic effect of the tick and bacterial transmission. By comparison, PE analysis of genes upregulated during transmission of *A. phagocytophilum* solely identified an over-representation of interferon signaling genes, interleukin-10 signaling, and other immune response pathways (Appendix A), confirming the effect of the bacterial transmission on these signaling pathways. In the case of the DEGs identified as upregulated during the tick bite only, no significant enrichment of pathways was detected (Appendix A).

In the case of shared downregulated genes, no significant PE was observed (Table 6), suggesting that the transmission of *A. phagocytophilum* leads to the downregulation of distinct pathways when compared with tick feeding. The analysis of genes downregulated only during *A. phagocytophilum* transmission, identified pathways involved in muscle contraction, collagen related pathways, and extracellular matrix organization (Appendix A). As in the case of shared downregulated genes, the PE analysis of genes downregulated during uninfected tick feeding did not identify any significantly over-represented pathways (Appendix A). These results were similar to the analysis of all DEGs downregulated during tick feeding versus expression in intact skin (Appendix A). This confirmed that *A. phagocytophilum* transmission, and not the tick feeding, is responsible for the detected effects on ECM organization. Whether this effect is due to bacterial manipulation of cells in the skin or due to changes in tick saliva during infection with *A. phagocytophilum* remains to be determined.

### 3.4. Confirmation of Th1 Cytokines Upregulation and Downregulation of ECM Genes by qRT-PCR

The expression patterns of selected genes identified by RNAseq were confirmed by qRT-PCR. RNA was isolated from skin biopsies of the bite site of uninfected and *A. phagocytophilum* ticks and were normalized to the gene expression of intact skin from the same mice. Similar to the tick batches used for the infestation of mice during the RNAseq experiments, relative bacterial levels were highly variable (Appendix A). The upregulation of *Ifn-γ*, *stat2*, and *il1β* during *A. phagocytophilum* transmission were validated (Figure 5a–c). Although *Irf1* was slightly upregulated in the bite site of *A. phagocytophilum* ticks, this difference was not statistically significant (Figure 5d). *S100a8*, *Acan*, and *matn3* were upregulated in samples taken from uninfected tick bite sites (Figure 5e–g). *Acan* and *matn3* encode for proteins with roles in ECM integrity; thus, validating the upregulation of interferon and interleukin-1 mediated signaling related genes and the downregulation of genes involved in ECM integrity. Interestingly, although the relative expression of immune genes (*Infg*, *stat2*, and *il1β*) was significantly lower in intact skin when compared with the bite site of A. phagocytophilum-infected ticks, the expression of *Acan* and *matn3* was not significantly different (Appendix A), thus, confirming that expression of these ECM integrity genes is similar in intact skin and *A. phagocytophilum* transmission sites.

## 4. Discussion

Initial responses within the skin can ultimately define the outcome of infection by a vector-borne pathogen. Skin cells, including immune cells, keratinocytes, and endothelial cells, serve as the site of initial replication for vector-borne viruses, bacteria, and parasites [31,32,33,34,35]. Arthropod inoculation of some viruses can lead to increased severity of pathologies associated with infection [31]. Further, arthropod saliva influences responses beyond the skin. For example, inoculation of mosquito salivary gland extracts (SGE) enhanced the migration of neutrophils and dendritic cells into draining lymph nodes and boosted the pathogenesis of the virus during antibody-dependent enhancement of dengue [34]. The role of single salivary proteins in the stimulation of immune cells and the enhanced pathogenesis of disease has also been investigated in mosquito models. In Zika virus (ZIKV), neutrophil-stimulating factor 1 (NeSt1) activates neutrophils and leads to augmented early virus replication [36]. Similarly, proteins within *Lutzomia longipalpis* saliva affect neutrophil function, inducing macrophage migration into the bite site and increasing *Leishmania chagasi* and *Leishmania infantum* replication [32,33,35]. Thus, arthropod saliva and other vector-derived factors are important determinants of pathogenesis of vector-borne pathogens.

In the case of tick-borne pathogens, little is known of the cellular and immune factors that influence their establishment and how tick saliva and tick modulatory molecules affect pathogen replication and pathogenesis [37]. In the case of *Rickettsia parkeri,* although tick saliva and tick feeding increase skin pathology during bacterial infection, bacterial replication at the inoculation site was reduced in the presence of tick salivary components [38,39]. As in the case of mosquitoes, *Amblyomma maculatum* feeding resulted in variable infiltration of macrophages and neutrophils in rhesus monkeys’ skin, whereas bacterial inoculation at the tick feeding site led to marked neutrophil and macrophage translocation at the dermis [39]. *Ixodes scapularis* feeding also led to a significant increase in the number of neutrophils and macrophages in murine skin [25], which was consistent with previous transcriptional analysis of tick bite sites in murine models [40] and our results (Appendix A). According to our results, genes involved in neutrophil chemotaxis and degranulation were induced during *I. scapularis* feeding and *A. phagocytophilum* transmission (Figure 3 and Appendix A). This includes the upregulation of CXCL1 and CXCL2 and the receptors CCR5, CCR7, and CCRL2 (Appendix A), which are involved in the activation, migration, effector, and antigen presentation functions of neutrophils [41,42]. Pathways enrichment (PE) analysis of genes upregulated during tick feeding and *A. phagocytophilum* transmission when compared with intact skin showed that neutrophil degranulation pathways are enriched in both (Table 5), suggesting that this effect may be mainly in response to tick feeding. This is corroborated by the absence of enrichment during *A. phagocytophilum* transmission versus feeding by uninfected ticks (Table 3). Interestingly, early experiments on sheep demonstrated that *A. phagocytophilum* infection led to higher number of neutrophils in the skin in the presence or absence of tick feeding [17]; whether *A. phagocytophilum* alone triggers neutrophil chemotaxis to the skin remains to be determined. Studies in the 1970s suggested that neutrophils may be involved in the pathology of tick bites of ixodid ticks [43]; nevertheless, what function neutrophils play during *A. phagocytophilum* transmission remains unexplored.

Among other immune pathways affected by the transmission of *A. phagocytophilum* in the skin was an increased activation of genes involved in interferon signaling (Table 3 and Table 5 and Figure 6). Type I (17 members, including Ifn-α and Ifn-β) and type II (Ifn-γ) interferons are important against viral and bacterial infections [44,45]. Their interaction with receptors within the membrane of cells activate signaling cascades that lead to the expression of immune-related genes [44]. The importance of Ifn-γ signaling in the immune response against systemic *A. phagocytophilum* infection has been demonstrated in several studies [10,11,12,13,14]. However, the significance of these cytokines and signaling pathways during early infection is unknown. Type I and type II interferon signaling pathways confer resistance to *R. parkeri* infection in murine models [46]. Double knockout mice with mutations in the type I IFN receptor (*Ifnar1 −/−*) and Ifn-γ receptor (*Ifngr1 −/−*) resulted in eschar development in the skin and lethality after intradermal (i.d.) infection of as low as 10^2^ bacteria when compared with 10^7^ when injected intravenously (i.v.), suggesting that IFN signaling in the skin may be important for protective immunity [46,47]. Furthermore, infection and vaccination with mutant *R. parkeri* strains showed that *Ifnar1 −/−*; *Ifngr1 −/−* double knockout mice may be a good tool for the study of rickettsial pathogenesis, and as models to evaluate vaccine candidates [46]. As with SFG *Rickettsia*, *A. phagocytophilum* does not produce disease in mice. Given our results, and that bacterial numbers increase during systemic infection of *A. phagocytophilum* in Ifn-γ *−*/*−* mice [11], it is possible that these receptors are also important for initial immune responses in the skin. However, more studies are needed to understand the role of IFN signaling during bacterial establishment.

Unexpectedly, *A. phagocytophilum* transmission also resulted in the downregulation of extracellular matrix (ECM) organization related genes (Table 4). Several genes encoding ECM structural components, such as collagen, thrombospondin-5, integrin ITGA10, Matrilin 3, aggrecan, elastin, fibronectin, and fibrillin, were downregulated during *A. phagocytophilum* transmission by unknown mechanisms (Figure 5; Appendix A). Enrichment of genes involved in collagen-related pathways was also observed when DEGs unique to *Anaplasma phagocytophilum* transmission, when compared with intact skin, were analyzed (Appendix A). A liver-derived protease—plasmin—degrades collagens, fibronectin, and other components of the ECM. Several pathogens, including *Borrelia* spp., exploit plasmin to facilitate their invasion into tissues [48]. Interestingly, α2-macroglobulin (A2m), a protein involved in the regulation of plasmin to avoid excessive proteolysis, is downregulated in the skin during *A. phagocytophilum* transmission (Appendix A). Whether this results in more damage in the ECM due to plasmin activity during *A. phagocytophilum* early infection remains to be determined. A second mechanism of ECM degradation that is manipulated by pathogens is the induction of host metalloproteinases [48]. *Anaplasma phagocytophilum* is known to stimulate the release of metalloproteinases during infection of neutrophils and during coinfection with *Borrelia burgdorferi* [49,50]. We detected the upregulation of several metalloproteinases in the site of *A. phagocytophilum* transmission, including MMP1b, MMP8, MMP13, and MMP25 (Appendix A). Both Granquist et al. [16] and Reppert et al. [17] reported the presence of infected neutrophils within the bite site of infected ticks and uninfected ticks on experimentally infected hosts, respectively. Although it is possible that infected neutrophils may be the source of this upregulation, the synergistic effect of tick saliva was also detected as these metalloproteinases were also upregulated in the tick-only control, except for MMP1b (Appendix A). Increased permeability of the ECM may facilitate the dissemination of the infection, warranting the further exploration of the molecular mechanisms and effects of the downregulation of ECM integrity genes.

## 5. Conclusions

Overall, our results indicate that *A. phagocytophilum* transmission leads to the activation of interferon signaling pathways and the upregulation of Th1 cytokines such as Tnfα, Ifn-γ, and other cytokines and chemokines associated with inflammation in the skin. The inflammatory environment developed in the skin, results in the activation of genes involved in neutrophil chemotaxis, confirming previously reported findings. Further, transmission of this tick-borne pathogen also causes a downregulation of genes involved in ECM integrity and an upregulation of metalloproteinases that may increase vascular leakage, indicating that *A. phagocytophilum* transmission and early infection actively delays wound healing responses and may affect vascular permeability at the bite site. Understanding the initial responses at the site of transmission of tick-borne pathogens can assist us to discover protective signaling pathways and identify models that can be used to distinguish factors that define disease pathogenesis and protective immunity, as previously shown with *R. parkeri* [46].

## Figures and Tables

**Figure 1 life-12-01965-f001:**
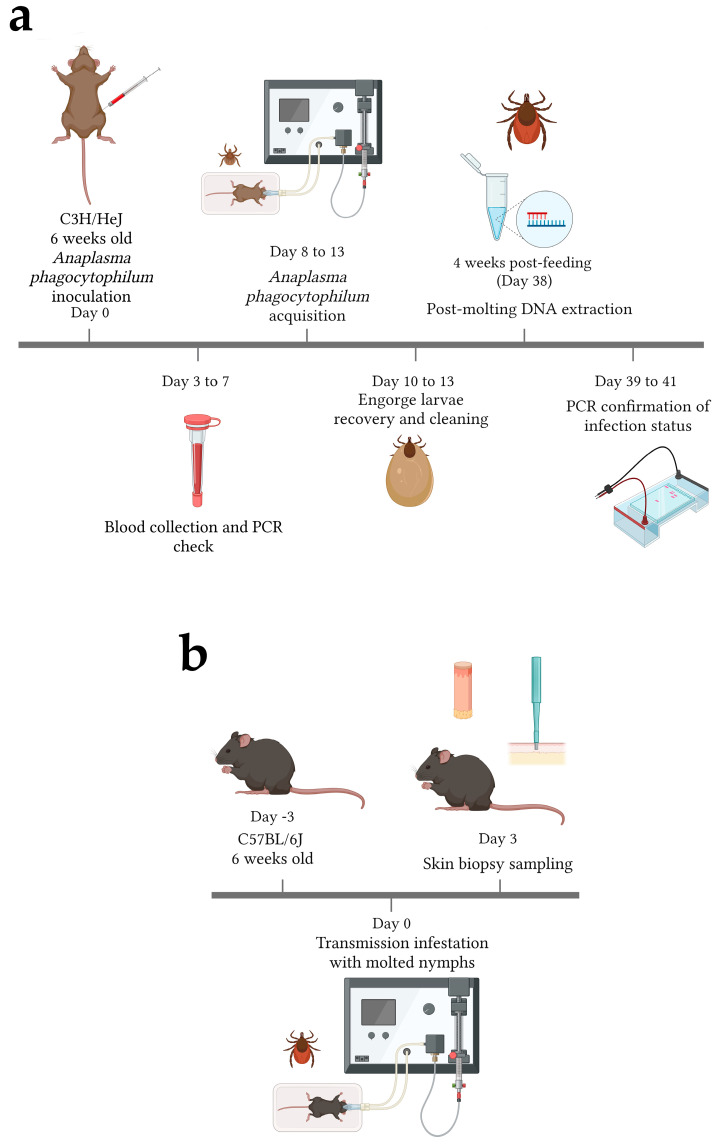
Schematic representation of acquisition and transmission feedings. (**a**) C3H/HeJ mice were inoculated with *Anaplasma phagocytophilum*-infected HL60 cells or PBS (control mice). Blood samples were taken on days 3, 5, and 7, and infection (or lack of) was confirmed by PCR. Animals were infested with pathogen-free certified larvae on day 8 once infectious status had been confirmed. Engorged larvae were recovered starting at day 10 (3 days after infestation in mice) until day 13 (5 days post infestation in mice). Larvae were allowed to molt for 1 month and nymphs were tested for *A. phagocytophilum* infection by PCR and gel electrophoresis. (**b**) After confirmation of infection (or lack of), nymphs were infested into C57BL/6J and were allowed to feed for 3 days. Skin biopsies were collected from the bite site of *Anaplasma phagocytophilum*-infected ticks, uninfected ticks, and from intact skin. Illustration was created using BioRender.

**Figure 2 life-12-01965-f002:**
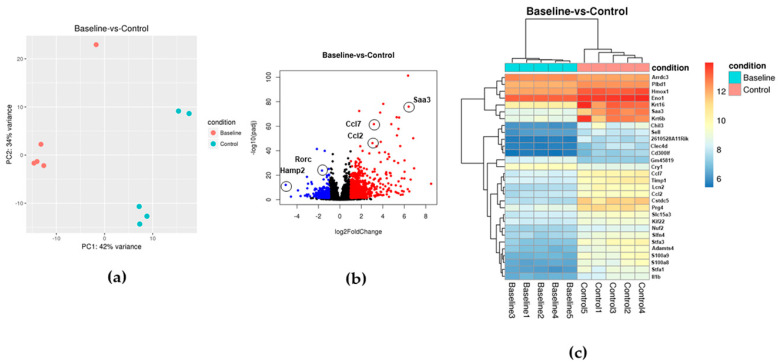
Transcriptional profile of immune genes during tick feeding. (**a**) Principal component analysis (PCA) of the transcriptional profiles of intact skin samples (baseline = pink dots) versus uninfected control ticks (blue dots). (**b**) The global transcriptional change across intact skin (baseline) and skin samples collected from uninfected tick bite sites (control) was visualized by a volcano plot. Each data point in the scatter plot represents a gene. The log2 fold change of each gene is represented on the *x*-axis and the log10 of its adjusted *p*-value is on the *y*-axis. Genes with an adjusted *p*-value less than 0.05 and a log2 fold change greater than 1 represent upregulated genes (red dots). Genes with an adjusted *p*-value less than 0.05 and a log2 fold change less than −1 are indicated by blue dots (downregulated genes). (**c**) A bi-clustering heatmap was used to visualize the expression profile of the top 30 differentially expressed genes sorted by their adjusted *p*-value by plotting their log2 transformed expression values in samples. The pink squares at the top represent “control” uninfected ticks, whereas the intact skin (baseline) is shown in light blue.

**Figure 3 life-12-01965-f003:**
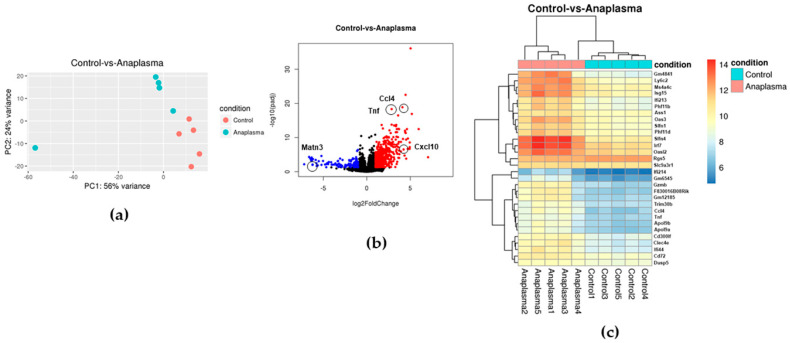
Transmission of *Anaplasma phagocytophilum* leads to Th1 and interferon signaling induction. (**a**) Principal component analysis (PCA) of the transcriptional profiles of skin samples from uninfected tick bite sites (control; pink dots) versus skin samples from infected tick bite sites (Anaplasma; blue dots). (**b**) The global transcriptional change across the skin samples collected from uninfected (control) and infected tick bite sites (Anaplasma) was visualized by a volcano plot. Each data point in the scatter plot represents a gene. The log2 fold change of each gene is represented on the *x*-axis and the log10 of its adjusted *p*-value is on the *y*-axis. Genes with an adjusted *p*-value less than 0.05 and a log2 fold change greater than 1 (upregulated) are indicated by red dots. Genes with an adjusted *p*-value less than 0.05 and a log2 fold change less than −1 (downregulated) are indicated by blue dots. (**c**) A bi-clustering heatmap was used to visualize the expression profile of the top 30 differentially expressed genes sorted by their adjusted *p*-value by plotting their log2 transformed expression values in samples. The pink squares at the top represent infected tick bite sites (Anaplasma). The light blue squares are the skin samples from uninfected tick’s bite sites (control).

**Figure 4 life-12-01965-f004:**
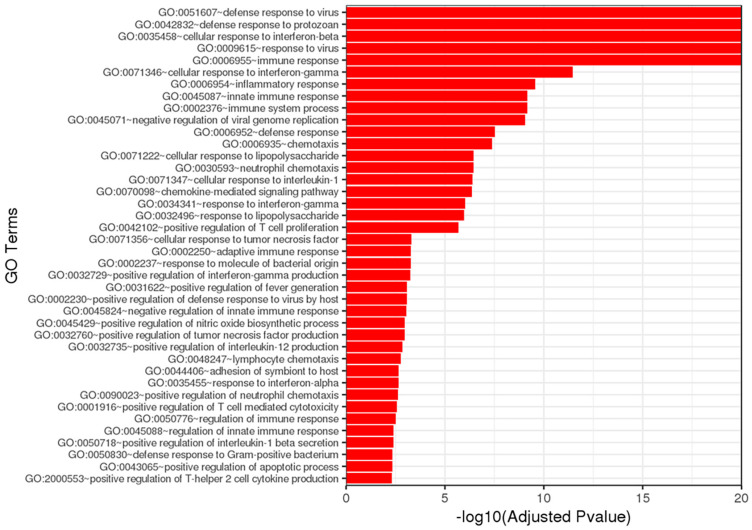
Gene ontology (GO) analysis of genes differentially expressed during transmission of *Anaplasma phagocytophilum*.

**Figure 5 life-12-01965-f005:**
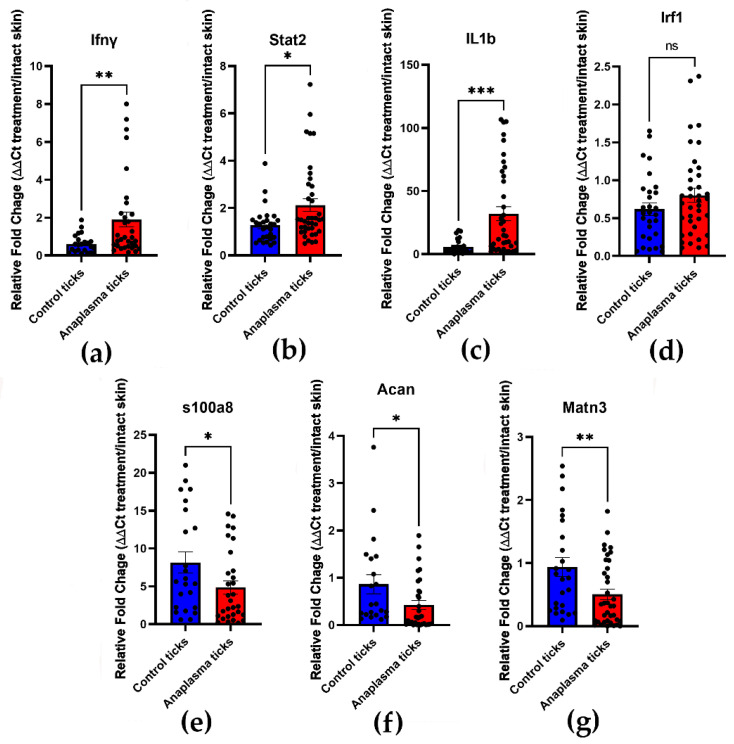
qRT-PCR confirmation of DEGs in skin samples taken from the bite site of uninfected and *Anaplasma phagocytophilum*-infected ticks. The ΔΔCt method was used to calculate the fold change gene expression on skin samples from uninfected tick bite sites (control; blue) versus *A. phagocytophilum*-infected (Anaplasma; red). Gene expression levels were normalized to actin and divided by normalized gene expression in intact skin. The expression of: (**a**) interferon γ (*Ifn-γ*; ** *p* = 0.0084); (**b**) signal transducer and activator of transcription 2 (*stat2*; ** p* = 0.0132); (**c**) interleukin β (*Il1β*; *** *p* = 0.0005); (**d**) interferon regulatory factor 1 (*Irf1*; ns: not significant); (**e**) S100 calcium binding protein A8 (*S100a8*; * *p* = 0.0412); (**f**) Aggrecan (*Acan*; * *p* = 0.0352); and (**g**) matn3 (*matn3*; ** *p* = 0.0091). Each dot in the graph represents a single skin sample. The height of the bar and the lines symbolize the average and the standard mean of the error (SEM), respectively. The graphs are representative from three independent experiments.

**Figure 6 life-12-01965-f006:**
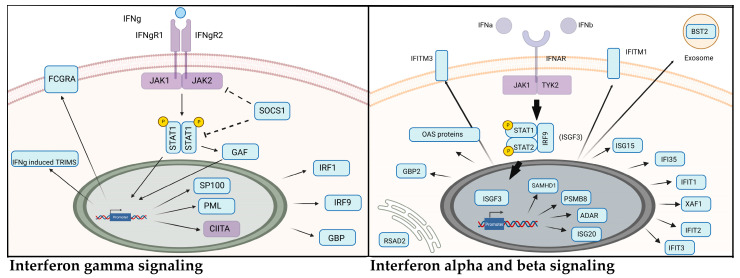
Interferon gamma and alpha/beta signaling related genes upregulated during *Anaplasma phagocytophilum* transmission. Components in light blue represent genes that are upregulated during transmission of *A. phagocytophilum* when compared with the bite site from uninfected ticks. Purple proteins are encoded by genes that were not differentially expressed. Figure created with BioRender based on pathway enrichment analysis from Reactome.

**Table 1 life-12-01965-t001:** Primers used for PCR and qRT-PCR amplifications.

Primer Name	Primer Sequence	Ta *	Product Size (bp)	Reference
Mactin F	5′-ACGCAGAGGGAAATCGTCCGTGAC-3′	60 °C	101	[24]
Mactin R	5′-ACGCGGGAGGAAGAGGATGCGGCAGTG-3′	60 °C		
Actin Is F	5′-GGTCATCACAATCGGCAA-3′	54 °C	108	[25]
Actin Is R	5′-ATGGAGTTGTACGTGGTCTC-3′	54 °C		
P44 F	5′-ATGGAAGGTAGTGTTGGTTATGGTATT-3′	56 °C	77	[26]
P44 R	5′-TTGGTCTTGAAGCGCTCGTA-3′	56 °C		
16s rRNA F	5′-GGTGAGTAATGCATAGGAATC-3′	53 °C	108	[27]
16s rRNA R	5′-GCTCATCTAATAGCGATAAATC-3′	53 °C		
rpoB F	5′-CTTTATCCTGCTTTAGAACAACATC-3′	52 °C	286	[18]
rpoB R	5′-GGTCCGTATGGTCTGGTTACT-3′	52 °C		
Ifng F	5′-AGCGTCATTGAATCACACCT-3′	54 °C	196	This study
Ifng R	5′-ATCAGCAGCGACTCCTTTTC-3′	54 °C		
IL1βF	5′-CCTGTGTAATGAAAGACGGC-3′	54 °C	216	This study
IL1βR	5′-TGTCCTGACCACTGTTGTTT-3′	54 °C		
Irf1 F	5′-ATAACTCCAGCACTGTCACC-3′	54 °C	177	This study
IrF1 R	5′-AAGGTCTTCGGCTATCTTCC-3′	54 °C		
Stat2 F	5′- TGGGACTTCGGCTTCTTGAC-3′	57 °C	247	This study
Stat2 R	5′- TCTTGGGATTTGGGCTGAGC-3′	57 °C		
S100a8 F	5′-CACCATGCCCTCTACAAGAA-3′	54 °C	161	This study
S100a8 R	5′-CCCACTTTTATCACCATCGC-3′	54 °C		
Acan F	5′-CAGATGGCACCCTCCGATAC-3′	57 °C	151	This study
Acan R	5′-GACACACCTCGGAAGCAGAA-3′	57 °C		
Matn3 F	5′-GAGGGTGGCTGTGGTGAACT-3′	59 °C	160	This study
Matn3 R	5′-GGCTTCCTCCATCGCTGTCT-3′	59 °C		

* Annealing temperatures.

**Table 2 life-12-01965-t002:** Significantly differentially regulated genes affected by tick feeding and *Anaplasma phagocytophilum* transmission.

Comparison	Upregulated Genes	Downregulated Genes	Total Significantly DEGs
Intact skin v uninfected tick bite sites	797	416	1213
Intact skin v Anaplasma-infected tick bite	1417	1142	2559
Uninfected tick bite sites v Anaplasma-infected tick bite	476	146	622

**Table 3 life-12-01965-t003:** Pathway enrichment (PE) analysis of upregulated genes during transmission of *Anaplasma phagocytophilum*.

Pathway Identifier	Pathway Name	Number Entities Found	Number Entities Total	Entities *p*-Value	Entities FDR
R-HSA-909733	Interferon alpha/beta signaling	50	188	1.11 × 10^−16^	1.52 × 10^−14^
R-HSA-6783783	Interleukin-10 signaling	38	86	1.11 × 10^−16^	1.52 × 10^−14^
R-HSA-913531	Interferon signaling	81	394	1.11 × 10^−16^	1.52 × 10^−14^
R-HSA-1280215	Cytokine signaling in immune system	149	1092	1.11 × 10^−16^	1.52 × 10^−14^
R-HSA-168256	Immune system	213	2684	1.11 × 10^−16^	1.52 × 10^−14^
R-HSA-449147	Signaling by interleukins	70	643	4.66 × 10^−15^	5.32 × 10^−13^
R-HSA-877300	Interferon gamma signaling	41	250	1.18 × 10^−14^	1.15 × 10^−12^
R-HSA-380108	Chemokine receptors bind chemokines	17	57	1.40 × 10^−10^	1.21 × 10^−8^
R-HSA-6785807	Interleukin-4 and interleukin-13 signaling	26	211	2.48 × 10^−7^	1.88 × 10^−5^
R-HSA-1169410	Antiviral mechanism by IFN-stimulated genes	14	94	1.97 × 10^−5^	0.001342146
R-HSA-375276	Peptide ligand-binding receptors	20	203	1.29 × 10^−4^	0.007973888
R-HSA-1169408	ISG15 antiviral mechanism	11	83	3.99 × 10^−4^	0.022716167
R-HSA-9705462	Inactivation of CSF3 (G-CSF) signaling	6	27	6.44 × 10^−4^	0.03349729

**Table 4 life-12-01965-t004:** Pathway enrichment (PE) analysis of downregulated genes during transmission of *Anaplasma phagocytophilum*.

Pathway Identifier	Pathway Name	Number Entities Found	Number Entities Total	Entities *p*-Value	Entities FDR
R-HSA-1474244	Extracellular matrix organization	29	329	1.11 × 10^−16^	3.26 × 10^−14^
R-HSA-2022090	Assembly of collagen fibrils and other multimeric structures	13	67	7.42 × 10^−13^	5.98 × 10^−11^
R-HSA-1474290	Collagen formation	15	104	8.05 × 10^−13^	5.98 × 10^−11^
R-HSA-1474228	Degradation of the extracellular matrix	17	148	8.20 × 10^−13^	5.98 × 10^−11^
R-HSA-8948216	Collagen chain trimerization	11	44	3.13 × 10^−12^	1.72 × 10^−10^
R-HSA-1650814	Collagen biosynthesis and modifying enzymes	13	76	3.52 × 10^−12^	1.72 × 10^−10^
R-HSA-3000178		13	79	5.66 × 10^−12^	2.38 × 10^−10^
R-HSA-216083	Integrin cell surface interactions	11	86	3.39 × 10^−9^	1.22 × 10^−7^
R-HSA-1442490	Collagen degradation	10	69	5.56 × 10^−9^	1.78 × 10^−7^
R-HSA-419037	NCAM1 interactions	6	44	9.59 × 10^−6^	2.78 × 10^−4^
R-HSA-1566948	Elastic fiber formation	6	46	1.23 × 10^−5^	3.20 × 10^−4^
R-HSA-8874081	MET activates PTK2 signaling	5	32	2.86 × 10^−5^	6.86 × 10^−4^
R-HSA-3000171	Non-integrin membrane-ECM interactions	6	61	5.86 × 10^−5^	0.001289879
R-HSA-375165	NCAM signaling for neurite outgrowth	6	70	1.24 × 10^−4^	0.002108039
R-HSA-186797	Signaling by PDGF	6	70	1.24 × 10^−4^	0.002108039
R-HSA-3656244	Defective B4GALT1 causes B4GALT1-CDG (CDG-2d)	3	9	1.38 × 10^−4^	0.002108039
R-HSA-3656225	Defective CHST6 causes MCDC1	3	9	1.38 × 10^−4^	0.002108039
R-HSA-3656243	Defective ST3GAL3 causes MCT12 and EIEE15	3	9	1.38 × 10^−4^	0.002108039
R-HSA-8875878	MET promotes cell motility	5	45	1.41 × 10^−4^	0.002108039
R-HSA-2022854	Keratan sulfate biosynthesis	4	37	7.38 × 10^−4^	0.010327613
R-HSA-2129379	Molecules associated with elastic fibers	4	38	8.14 × 10^−4^	0.011395335
R-HSA-2022857	Keratan sulfate degradation	3	22	0.001819501	0.023653508
R-HSA-2243919	Crosslinking of collagen fibrils	3	24	0.002325531	0.027906369
R-HSA-1638074	Keratan sulfate/keratin metabolism	4	52	0.002542129	0.029455221
R-HSA-399710	Activation of AMPA receptors	2	7	0.002677747	0.029455221
R-HSA-6806834	Signaling by MET	5	88	0.002781469	0.03059616
R-HSA-1369062	ABC transporters in lipid homeostasis	3	29	0.00394569	0.039456898
R-HSA-8951671	RUNX3 regulates YAP1-mediated transcription	2	9	0.004364679	0.043646794

**Table 5 life-12-01965-t005:** Pathway enrichment (PE) analysis of upregulated genes shared between skin samples from uninfected tick bite sites and infected tick bite sites when compared with intact skin samples.

Pathway Identifier	Pathway Name	Number Entities Found	Number Entities Total	Entities *p*-Value	Entities FDR
R-HSA-6783783	Interleukin-10 signaling	36	86	7.12 × 10^−14^	1.17 × 10^−10^
R-HSA-6798695	Neutrophil degranulation	61	480	4.41 × 10^−13^	3.62 × 10^−10^
R-HSA-380108	Chemokine receptors bind chemokines	24	57	7.27 × 10^−11^	3.97 × 10^−8^
R-HSA-6785807	Interleukin-4 and interleukin-13 signaling	42	211	2.18 × 10^−8^	8.94 × 10^−6^
R-HSA-2500257	Resolution of sister chromatid cohesion	28	134	4.57 × 10^−7^	1.50 × 10^−4^
R-HSA-2467813	Separation of sister chromatids	27	195	1.10 × 10^−6^	3.00 × 10^−4^
R-HSA-141424	Amplification of signal from the kinetochores	22	94	7.82 × 10^−6^	0.001602622
R-HSA-141444	Amplification of signal from unattached kinetochores via a MAD2 inhibitory signal	22	94	7.82 × 10^−6^	0.001602622
R-HSA-5663220	RHO GTPases activate formins	24	149	1.20 × 10^−5^	0.002177163
R-HSA-68877	Mitotic prometaphase	31	211	1.79 × 10^−5^	0.002933421
R-HSA-9648025	EML4 and NUDC in mitotic spindle formation	24	121	3.02 × 10^−5^	0.004498095
R-HSA-69618	Mitotic spindle checkpoint	22	111	1.34 × 10^−4^	0.018203955

**Table 6 life-12-01965-t006:** Pathway enrichment (PE) analysis of downregulated genes shared between skin samples from uninfected tick bite sites and infected tick bite sites when compared with intact skin samples.

Pathway Identifier	Pathway Name	Number Entities Found	Number Entities Total	Entities *p*-Value	Entities FDR
R-HSA-400253	Circadian clock	9	105	0.001723	0.60096047
R-HSA-5682910	LGI-ADAM interactions	3	14	0.005775	0.60096047
R-HSA-3000480	Scavenging by Class A receptors	5	49	0.008965	0.60096047
R-HSA-8874081	MET activates PTK2 signaling	4	32	0.009601	0.60096047
R-HSA-3000178	ECM proteoglycans	6	79	0.016775	0.60096047
R-HSA-2022870	Chondroitin sulfate biosynthesis	3	25	0.026832	0.60096047
R-HSA-1482922	Acyl chain remodeling of PI	3	25	0.026832	0.60096047
R-HSA-391903	Eicosanoid ligand-binding receptors	3	25	0.026832	0.60096047
R-HSA-8949275	RUNX3 regulates immune response and cell migration	2	10	0.02741	0.60096047
R-HSA-8875878	MET promotes cell motility	4	45	0.029135	0.60096047
R-HSA-1482925	Acyl chain remodeling of PG	3	26	0.029633	0.60096047
R-HSA-1442490	Collagen degradation	5	69	0.033148	0.60096047
R-HSA-430116	GP1b-IX-V activation signaling	2	12	0.038188	0.60096047
R-HSA-3000170	Syndecan interactions	3	29	0.038924	0.60096047
R-HSA-1482801	Acyl chain remodeling of PS	3	31	0.045843	0.60096047
R-HSA-6785807	Interleukin-4 and interleukin-13 signaling	10	211	0.046493	0.60096047
R-HSA-1650814	Collagen biosynthesis and modifying enzymes	5	76	0.046724	0.60096047
R-HSA-391908	Prostanoid ligand receptors	2	14	0.050296	0.60096047
R-HSA-2214320	Anchoring fibril formation	2	15	0.0568	0.60096047

## Data Availability

RNAseq raw sequences were deposited into NCBI with accession # as follows: Intact skin (Baseline) 1 = SRR18095180; Intact skin (Baseline) 2 = SRR18097073; Intact skin (Baseline) 3 = SRR18105828; Intact skin (Baseline) 4 = SRR18106047; Intact skin (Baseline) 5 = SRR18106143; Uninfected tick bite site (Control) 1 = SRR18106222; Uninfected tick bite site (Control) 2 = SRR18147549; Uninfected tick bite site (Control) 3 = SRR18147908; Uninfected tick bite site (Control) 4 = SRR18148061; Uninfected tick bite site (Control) 5 = SRR18148857; Anaplasma 1 = SRR18150648; Anaplasma 2 = SRR18150707; Anaplasma 3 = SRR18150718; Anaplasma 4 = SRR18151819; and Anaplasma 5 = SRR18153448. The sanger sequences from the qRT-PCR primers were deposited into NCBI under the following accession numbers: Ifng = OM427501; Il1b = OL792765; Irf1 = OL792763; Stat2 = OL792766; S100a8 = OL792762; Acan = OM427502; Matn3 = OM628810.

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
