# Peer review of "Anaplasma phagocytophilum Transmission Activates Immune Pathways While Repressing Wound Healing in the Skin"

_life, 2022, doi:10.3390/life12121965_

Round 1
Reviewer 1 Report
Simple summary and abstract of the manuscript are well-organized and help the reader to grasp the problem, authors’ questions, findings, and methods.
The methods and objectives are clearly articulated. A few questions I have are listed below.
1. Why there are two types of mice used in the study? IF C3H/HeJ mice have high susceptibility why do authors switch to C57BL/6J mice?
2. What is the age and sex of the animals?
3. A schematic representation showing the different mice, and timelines would be a helpful visualization to understand the experiment design.
Data presented in the results section matches the analysis plan and they are extensively analyzed.
1. I was wondering if there is any quantitative data on the bacteria titers both for the mice infected initially for the pathogen acquisition and from the samples from mice infested with those ticks.
2. Are there any skin samples processed for histology?
3. The samples are grouped as the baseline, control, and Anaplasma. When you read the whole paper, the figures are clear. For the readers to have a sense of the sample set and for easier comparison, could the groups be named differently? Like intact skin, uninfected tick bite site, and infected tick bite site. This is just a suggestion.
4. If that is not too much work, in Figure 4 adding labels to each figure (like Ifnγ, stat2) would also make it easier for readers to compare the data at a glance.
5. If I am not wrong the sentence in Line 315 is missing ‘downregulated’.
6. In the section analyzing data from Anaplasma-infected and uninfected tick bite sites and DEGs, a little bit more comparison of numbers would be nice (the shared DEGs that might be attributed to the tick bite itself that authors observed in both cases).
7. In the analysis of the results in lines 318-325, the bacteria titers of the infected ticks and skin samples could be helpful since the authors also speculated that the outlier might have a lower level of infection. In any case, it would be nice to know and show that these mice received a similar number of bacteria during the infestation.
Author Response
We thank the reviewers for their valuable comments. The manuscript has been improved after incorporating their recommendations. All the changes have been highlighted using track changes and are mentioned herein.
Please find our response to each reviewer below:
Reviewer 1
Simple summary and abstract of the manuscript are well-organized and help the reader to grasp the problem, authors’ questions, findings, and methods.
The methods and objectives are clearly articulated. A few questions I have are listed below.
- Why there are two types of mice used in the study? IF C3H/HeJ mice have high susceptibility why do authors switch to C57BL/6J mice?
Response: We appreciate the question from reviewer 1. Here, we will explain our rationale for the use of two mice strains. Although the C3H/HeJ strain of mice is in fact highly susceptible to gram-negative bacterial infections and have been used to study the pathology of Anaplasma phagocytophilum infection. Poltorak et al. 1998 (Science) showed that this susceptibility is connected to the mutation of the gene encoding Toll-like receptor 4, which acts in the signal transduction of immune and inflammatory responses. Given that this mutation would likely affect signaling pathways important for the response to Anaplasma phagocytophilum transmission, we decided to use a mouse strain that has been previously used to study systemic immune responses to A. phagocytophilum (B6 mice; Martin et al. 2001). tlr4 was not among the differentially regulated genes in the skin between any of the conditions (intact skin versus tick bite, intact skin versus A. phagocytophilum infected tick bite, or control tick bite versus A. phagocytophilum infected tick bite). However, we did not have this information at the moment of the experimental design and were afraid of any possible impact on our results. Further, since the goal of the feeding on C3H is only to obtain infected ticks, any effect of the tlr4 mutation on immune responses would not affect the main objective of the second set of experiments (to define the transcriptional changes in immunocompetent mice).
We have included our reasoning for the use of both strains in the material and methods as follow:
Page 3 lanes 132 to 137: “C3H/HeJ male mice of 6 weeks old (The Jackson Laboratory, Bar Harbor, ME USA) were used for pathogen acquisition due to their high susceptibility to infection from gram negative bacteria, including to A. phagocytophilum infection [19]. The susceptibility of this mice strain to gram-negative bacteria is associated with a mutation in the cytoplasmic domain of Toll like receptor 4 (TLR4) [20] and has shown impaired inflammatory and innate immune responses under several conditions [21,22].”
Page 5 lanes 205 to 209: “Due to the potential effect of the mutation of C3H/HeJ mice tlr4 in the local immune responses to A. phagocytophilum transmission, we decided to use a different mice strain. The C57BL/6J mice strain has been previously used for the study murine systemic immune responses during A. phagocytophilum infection and the role of IFN-γ/STAT1 [11], therefore we used this same strain to define local immune responses to bacterial transmission.”
- What is the age and sex of the animals?
Response: We thank reviewer 1 for his/her question as it will help improve reproducibility. The required information has been added in line 132 (“C3H/HeJ male mice of 6 weeks old”) and line 211 (A. phagocytophilum infected and uninfected nymphs were used to infest 6 weeks old C57BL/6J male mice”).
We only used male mice as female mice are better at grooming and less ticks are recovered/bite when female mice are used, which is an observation after several years of experience working with mice models and tick feeding at the University of Minnesota and University of Maryland.
- A schematic representation showing the different mice, and timelines would be a helpful visualization to understand the experiment design.
Response: We appreciate the reviewers suggestion. We have added a schematic representation of the experiment with a timeline of each step. This is now “Figure 1” found in page 5. We have added reference to the figure throughout the methods section. The rest of the figure numbers has been changed accordingly.
Data presented in the results section matches the analysis plan and they are extensively analyzed.
- I was wondering if there is any quantitative data on the bacteria titers both for the mice infected initially for the pathogen acquisition and from the samples from mice infested with those ticks.
Response: There is no quantitative data of the bacterial titers in the mice. Infection status was only confirmed by regular PCR as described in the materials and methods. DNA is still available to perform this quantification. In a recent paper by Urbanova et al. published in pathogens title “Experimental Infection of Mice and Ticks with the Human Isolate of Anaplasma phagocytophilum NY-18” demonstrated that A. phagocytophilum infection rates decreased from acquisition (56%) until molting (15.7%), despite infesting at the peak of mouse infection in day 8. This has been our experience in several of our previous studies. Thus, molting is the bottleneck of infection in ticks. Therefore, we decided focus in the molted nymphs for the quantification experiments requested. The relative numbers of bacteria in the ticks are now reported in Figure S1 and discussed more in detail in our answer to reviewer’s 1 question.
- Are there any skin samples processed for histology?
Response: No histology samples were taken at this time. We are following up with this project with histology, immunohistochemistry, and flow cytometry to evaluate differences in neutrophil migration. However, we limited ourselves to the gene expression analysis for this study.
- The samples are grouped as the baseline, control, and Anaplasma. When you read the whole paper, the figures are clear. For the readers to have a sense of the sample set and for easier comparison, could the groups be named differently? Like intact skin, uninfected tick bite site, and infected tick bite site. This is just a suggestion.
Response: We appreciate Reviewers 1 suggestion. We have changes to “intact skin” throughout the manuscript and have made the indication that it refers to the “baseline” in the figures. The sequences have been labeled as “Baseline” at the time of the submission to GENEWIZ and all associated files are labeled as such. Since they have been already submitted to NCBI, we prefer to maintain the figures labeled as “baseline”, “control”, and “Anaplasma” to make it easier for other members of the scientific community to reanalyze our data if desired. We have made the change in figure 5 (previously 4) presenting the qRT-PCRs run in house.
- If that is not too much work, in Figure 4 adding labels to each figure (like Ifnγ, stat2) would also make it easier for readers to compare the data at a glance.
Response: We agree with the reviewer that the data will be easier to compare when labels are added to each figure. We have added the labels in now Figure 5 as requested.
- If I am not wrong the sentence in Line 315 is missing ‘downregulated’.
Response: We thank the reviewer for point out our mistake. We had in fact omitted the word downregulated. The sentence now reads “This encompassed 476 upregulated and 146 downregulated genes, indicating a synergistic effect between A. phagocytophilum and the tick (Table 2).”
- In the section analyzing data from Anaplasma-infected and uninfected tick bite sites and DEGs, a little bit more comparison of numbers would be nice (the shared DEGs that might be attributed to the tick bite itself that authors observed in both cases).
Response: We have added the analysis requested by the reviewers in a new section “3.3 Differentially expressed genes (DEGs) stimulated during tick feeding and A. phagocytophilum transmission” in page 14 lines 437 to 472 and includes the addition of two new tables (Table 5 and 6) in the main manuscript, 4 supplemental tables (Table S5-S8), and one supplementary file (supplementary file S7).
We have added the discussion of these results in page 18 lanes 537 to 542 as follow: “Pathways enrichment (PE) analysis of gene upregulated during tick feeding and A. phagocytophilum transmission when compared to intact skin showed that neutrophil degranulation pathways are enriched in both (Table 5), suggesting that this effect may be mainly in response to tick feeding. This is corroborated by the absence of enrichment in during A. phagocytophilum transmission versus feeding by uninfected ticks (Table 3).” and in page 18 lanes 581 -583 as follow: “Enrichment of genes involved in collagen related pathways was also observed when DEGs unique to Anaplasma phagocytophilum transmission when compared to intact skin were analyzed (Table S7). “
- In the analysis of the results in lines 318-325, the bacteria titers of the infected ticks and skin samples could be helpful since the authors also speculated that the outlier might have a lower level of infection. In any case, it would be nice to know and show that these mice received a similar number of bacteria during the infestation.
Response: Unfortunately, we do not have access to the skin RNA submitted to GENEWIZ as we did not request for its return. Thus, we cannot determine the bacterial numbers in the skin. We have added the following statement: “although this is speculative since we did not test bacterial numbers in the skin” in lane 365. We have, however, performed qPCR analysis of the relative bacterial levels in each of the tick batches originally tested by regular PCR. As expected, the levels of infection varied between samples. We have added the quantification of the batches used for the RNAseq infestation (Figure S1c) and representative batches for the qRT-PCR validation experiments (Figure S1d).
The materials and methods have been updated and now described procedure in page 6 lines 195 – 203 as follow:
“Additionally, the relative levels of bacterial infection were assessed by qPCR, using the ΔCt value of A. phagocytophilum p44 normalized by tick actin with the following formula:
qPCRs were performed using PowerUp™ SYBR™ Green Master Mix (Applied Biosystems, Whaltham, MA), using same primers as for PCR analysis (Table 1). Amplification, melt curves, and data were analyzed with the CFX Maestro Software (Bio-Rad, Hercules, CA).”
The results have been modified in page 9 lane 345 as follows: “Ticks were allowed to molt and DNA was extracted from 5 nymphs from each group to confirm infection status. Anaplasma phagocytophilum positive nymphs (Figure S1b) with varying relative levels of bacterial infection (Figure S1c) were infested onto naïve C57BL/6J mice for 3 days.” And in page 15 lanes 479 to 481: “Similar to the tick batches used for the infestation of mice during the RNAseq experiments, relative bacterial levels were highly variable (Figure S1d).”.

Reviewer 2 Report
The manuscript is well designed, experiments are performed with modern methods, the results are well presented and discussed.
Author Response
We thank the reviewers for their valuable comments. The manuscript has been improved after incorporating their recommendations. All the changes have been highlighted using track changes and are mentioned herein.
Please find our response to each reviewer below:
Reviewer 2
The manuscript is well designed, experiments are performed with modern methods, the results are well presented and discussed.
Response: We appreciate the comments from the reviewer. We are glad that the reviewer agrees with our methodology and how we presented/discussed our results.
